# Modulating Band Gap of Boron Doping in Amorphous Carbon Nano-Film

**DOI:** 10.3390/ma12111780

**Published:** 2019-05-31

**Authors:** Rui Zhu, Qiang Tao, Min Lian, Xiaokang Feng, Jiayu Liu, Meiyan Ye, Xin Wang, Shushan Dong, Tian Cui, Pinwen Zhu

**Affiliations:** State Key Laboratory of Superhard Materials, College of Physics, Jilin University, 2699 Qianjin Street, Changchun 130012, China; ruizhu16@mails.jlu.edu.cn (R.Z.); lianmin16@mails.jlu.edu.cn (M.L.); Fengxk16@mails.jlu.edu.cn (X.F.); 18743415130@sina.cn (J.L.); 17808074803@163.com (M.Y.); xin_wang@jlu.edu.cn (X.W.); dongss@jlu.edu.cn (S.D.); cuitian@jlu.edu.cn (T.C.)

**Keywords:** semiconductor, amorphous materials, carbon materials

## Abstract

Amorphous carbon (a-C) films are attracting considerable attention to due their large optical band gap (E_opt_) range of 1–4 eV. But the hopping conducting mechanism of boron doping a-C (a-C:B) is still mysterious. To exploring the intrinsic reasons behind the semiconductor properties of a-C:B, in this work, the boron doping a-C (a-C:B) nano-film was prepared, and the growth rate and E_opt_ changing were analyzed by controlling the different experimental conditions of magnetron sputtering. A rapid deposition rate of 10.55 nm/min was obtained. The E_opt_ is reduced from 3.19 eV to 2.78 eV by improving the substrate temperature and sputtering power. The proportion of sp^2^/sp^3^ increasing was uncovered with narrowing the E_opt_. The shrinking E_opt_ can be attributed to the fact that boron atoms act as a fluxing agent to promote carbon atoms to form sp^2^ hybridization at low energy. Furthermore, boron atoms can impede the formation of σ bonds in carbon atom sp^3^ hybridization by forming B–C bonds with high energy, and induce the sp^3^ hybridization transfer to sp^2^ hybridization. This work is significant to further study of amorphous semiconductor films.

## 1. Introduction

Semiconductor films are the most common form of amorphous carbon (a-C) films, owing to the superior mechanical, optical, and opto-electronic properties [1,2,3,4,5]. Even though the carbon atoms are disordered in a-C, the optical band gap (E_opt_) is still controllable, and the E_opt_ can be changed over a quite large range from 1 eV to 4 eV [6,7]. Thus, the superior properties of a-C arise from adjustable E_opt_, which can extend to visible light [8,9]. This is an effective way to adjust the E_opt_ with doping boron, nitrogen and phosphorus [10], which can form n-type or p-type semiconductors [11,12]. For example, nitrogen/phosphorus incorporate a-C is n-type semiconductor, and E_opt_ decreases from 2.5 eV to 1.6 eV with increasing nitrogen content [11]. The reason for the change in E_opt_ is attributed to modifications in the proportion of sp^2^ and sp^3^ hybridization [13,14]. A high content of sp^3^ hybridization can induce diamond-like properties, such as tetrahedral a-C (ta-C) [15], and the sp^2^ hybridization of a-C can generate graphite-like properties. Carey et al. described a-C as a conductive sp^2^ phase embedded in electrically insulating sp^3^ matrix [16]. Furthermore, a-C demonstrates hopping conductivity; hence, a high proportion of sp^2^ hybridization is preferred to narrow the E_opt_ [17]. However, the reasons why doping elements interconvert between sp^2^ and sp^3^ hybridization are still unclear, and further study to uncover the basis of the semiconductor properties of a-C is required.

Boron has one fewer electrons than carbon, which allows it to generate p-type doping in a-C [7,18]. Boron atoms can release stress with a high sp^3^ content in boron-incorporated ta-C (ta-C:B) [15,19], and boron atoms can passivate the dangling bonds on the surface of carbon atoms, which can enhance the stability of materials to resist oxidization and corrosion [11]. In general, boron doping in a-C negatively affects conductance due to the presence of boron atoms which are void of π electrons, which is detrimental to sp^2^ hybridization. Schenk et al. among others found that sp^3^ hybridization becomes dominated with increasing boron concentration in boron doping hydrogenated a-C film [20]. Nastasi et al. also showed that boron doping can increase sp^3^ hybridization and C–H bonds in hydrogenated a-C [21]. However, in most cases, the sp^3^ hybridization decreases with increasing boron content. For example, Tan et al. found that sp^3^ content decreased from 73.8% to 42.5% with boron concentrate increasing from 0.59% to 6.04% in boron-incorporated a-C (a-C:B) [22]. Meanwhile, Rusop et al. reported that the E_opt_ of a-C:B was almost unchanged with a boron content below 10%, but that E_opt_ decreased with higher boron content, i.e., more than 10%, and they considered that the decreased E_opt_ was not caused by an increased sp^2^ content [16]. Until now, the means by which doping boron influences the hybridization of carbon atoms in a-C remains a mystery. Further study of a-C:B films is important to discover how to modulate the E_opt_ of a-C, and to understand the inherent reasons behind the interconverted mechanism of sp^2^ and sp^3^ hybridization in a-C:B.

In this work, a-C:B nano-films were prepared by magnetron sputtering. The most superior growth rate was determined to obtain a maximum thickness of 5 μm, and the E_opt_ was analyzed at a fast growth rate. The interconvert between sp^2^ and sp^3^ in a-C:B films were analyzed under different experimental conditions of increasing boron target power, carbon target power, substrate temperature, and working pressure. Finally, a reason for the doping boron atoms inducing to form sp^2^ hybridization is put forward. This work is significant in the development of new functional amorphous semiconductor films.

## 2. Experimental

Magnetron Sputtering (JGP450, Shenyang Qihui Vacuum Technology Co., Ltd., Shenyang, China) was used to prepare the a-C:B films. The substrate material was glass, and both the boron and carbon targets were pure to 99.999%. The distance between the substrate and target was 55 mm in all cases. The working gas was Ar. Synthesized films were tested by X-ray photoelectron spectroscopy (XPS using a Surface Science Instruments spectrometer, ESCALAB 250, Thermo Electron Corporation, MA, USA) with a focused Al Kα radiation (1486.6 eV), scanning electron microscopy (SEM, MAGELLAN 400, FEI, Hillsboro, OR, USA). High resolution transmission electron microscopy (HRTEM JSM-2100FS, Tokyo, Japan, equipped with selected area electron diffraction (SAED)) was performed to determine the morphology. Raman spectra were studied on a Renishaw (Shanghai, China) in Via micro-Raman spectrometer at room temperature with a 532 nm laser. The thickness of films was analyzed by atomic-Profiler (Veeco Dektak 150, Veeco, Shanghai, China) to determine the deposition rate. Optical transmittance spectra were measured with a UV-3150 double-beam spectrophotometer (Shimadzu, Japan, Japan) in the range from 300 nm to 900 nm. The E_opt_ for a-C:B films was obtained via the Tauc relationship, (αhν)^2^ = c(hν−E_opt_), where α is the absorption coefficient, c is a constant, and hν is photon energy [23].

## 3. Results and Discussion

The deposition rate related to the experimental conditions is shown in Figure 1. The deposition rate of the films increased at first and decreased later with increasing the working pressure or substrate temperature (Figure 1a). The fast deposition rate was 10.55 nm/min at 200 °C and 1.0 Pa. Higher working pressure decreases the mean free path of argon molecules. So, the frequency of the sputtered atoms (boron atoms and carbon atoms) crashing with argon molecules was increased, the emissive power of secondary electrons become stronger, the sputtering capacity also be increased, and thus, the deposition rate was raised. The frequency of sputtered atoms crashing with argon molecules will greatly increase with a working pressure greater than 1.0 Pa, but it is detrimental to the energy of the sputtered atoms due to their dramatic crashes with argon molecules. The quantity of boron and carbon atoms reaching the substrate decreased; thus, the deposition rate decreases with a working pressure higher than 1.0 Pa. In Figure 1a, the reaction rate of boron and carbon atoms is improved with a higher substrate temperature, i.e., of 200 °C, rather than 100 °C. Therefore, the deposition rate increased. Meanwhile, the migration rate of the deposited atoms on the substrate increased when the substrate temperature was higher than 200 °C, which will yield dense films. Hence, the deposition rate decreased with higher substrate temperature, i.e., 200–400 °C; So, a fast deposition rate of 200 °C and 1.0 Pa were chosen in below experiments.

The deposition rate relates to the sputtering power of boron and carbon targets, indicating an undulant curve (Figure 1b). The sputtering rate of boron and carbon atoms will increase with increasing the sputtering power, which may serve to improve the deposition rate. However, by further increasing the sputtering power, the deposition rate decreased to increase the re-sputtering rate of the deposition atoms. Thus, the undulance of the deposition rate at high power may be attributed to the balance of the sputtering rate and the re-sputtering rate of the deposition atoms. The optimum sputtering power for boron is 160 W, and for carbon is 50 W. These settings were chosen in after performing the experiments. The thickness of the film can reach 5 µm with 10 h deposition under optimal conditions.

According to HRTEM results (Figure 2a), the synthesized samples were amorphous. The amorphous nature of the samples was also confirmed by SAED (insert pattern in Figure 2a). The samples are disordered but there was less B_2_O_3_ on the surface, which is due to surface oxidation. The morphology of a-C:B films is granulated with a nano size of about 30–80 nm in SEM results (Figure 2b). Thus, the synthesized samples are nano amorphous films.

As synthesized a-C:B was investigated by UV-vis diffuse reflectance spectroscopy, and the E_opt_ was obtained using the Tauc equation shown in Table 1 and Figure 3. By increasing the working pressure from 1.0 Pa to 2.0 Pa, E_opt_ reduced from 3.19 eV to 3.08 eV (Table 1, sample 1,5). The reason for this may be the higher pressure, i.e., 2.0 Pa, enhancing the interaction between the boron and carbon atoms. In Table 1, samples 1,6,7, the E_opt_ reduce just a little (3.19 eV to 3.15 eV) with increasing the temperatures from 200 °C to 400 °C. Thus, temperature may not directly influence the electronic structure, in contrast to changing the content of defect or sp^2^ cluster size [16]. The E_opt_ narrowed from 3.19 eV to 2.90 eV (increasing carbon target power, Figure 3a), and from 3.19 eV to 2.78 eV (increasing boron target power, Figure 3b). When the boron target power reached 240 W, sample 3 indicated two E_opt_ of 2.85 eV and 2.78 eV. In Figure 3b, 3.19 eV, 3.03 eV, and 2.85 eV are compared with 3.19 eV, 3.05 eV, and 2.90 eV in different carbon target powers (Figure 3a). The 2.78 eV may arise from the high boron target power, which indicates that boron atoms have a different state at a power of 240 W. Thus, the target power may be an important factor in modulating the boron and carbon atom combination.

In order to understand the reason behind the E_opt_ change, the energy states of boron and carbon atoms were analyzed by XPS measurements under the highest deposition rate conditions; only a single condition was changed to study the influence (Table 1, Figure 4). At a lower pressure, i.e., 1.0 Pa, there are no clear B 1s band energy states (Table 1, sample 1); when increasing the pressure to 2.0 Pa, B 1s also does not change (Table 1, sample 5). Because the temperature is low (200 °C) and boron and carbon target powers are just 160 W and 50 W respectively, even with the working pressure increasing to 2.0 Pa, the lower energy can’t induce boron atoms to interact with a-C. But with lower energy (Table 1, sample 1,5), C 1s states can be fitted as four states. The high energy states, 286 eV and 288 eV, are attributable to oxidized carbon (C–O) [11,24], while 284.9 eV arises from C–C σ bonds of sp^3^ hybridization, which represent the properties of ta-C [14,22]. Meanwhile, the 284.1 eV is due to C=C, which is caused by sp^2^ hybridization [14,22]. Hence, a-C:B is composed of a sp^2^ carbon cluster and sp^3^ carbon matrix. With Raman results, the I_D_/I_G_ represents a proportion of sp^2^/sp^3^ [25], and the I_D_/I_G_ is not changed with increasing the pressure from 1 to 2 Pa; however, the E_opt_ is reduced from 3.19 eV to 3.08 eV. The reason for this may be that, although a higher working pressure, i.e., 2.0 Pa, can’t supply enough energy to induce the reaction between boron and carbon atoms, it may shape the size or vary the arrangement of sp^2^ clusters, which may narrow the hopping distance between two sp^2^ clusters.

Higher pressure may not directly provide higher energy for a-C:B. Increasing the temperature is effective to improve the reaction energy. As reported, temperature is a crucial factor in modulating the E_opt_ of a-C by varying defect content and the size of the sp^2^ cluster [16]. As shown in Table 1, samples 1,6,7, a higher substrate temperature increases the content of sp^2^, because I_D_/I_G_ increase from 57% to 61%. This is the reason for the narrowing of the E_opt_ from 3.19 eV to 3.15 eV. The oxidized boron (191.4 eV) appears at a high temperature, i.e., 300 °C (Table 1, sample 6,7), which is consistent with the HRTEM results of B_2_O_3_ on the surface of the samples. Moreover, higher temperatures induce binding energy states at B 1s 189.2 eV and C 1s 283.2 eV, which occurs due to B–C bonds in B_4_C (Table 1, sample 7) [14]. Thus, higher temperatures facilitate the combination of boron and carbon atoms, and the B–C bonds appear in XPS when the temperature reaches 400 °C. Finally, higher temperatures yield a greater content of sp^2^ and more B–C bonds.

B–C bonds also appear at a high boron target power, i.e., 240 W (Table 1, sample 3, Figure 4e). High boron target power means high energy with boron atoms, which can promote the reaction between boron and carbon atoms. In XPS, the proportion of C=C and C–C represents the proportion of sp^2^ and sp^3^, which relates to the electrical properties [26]. Thus, a higher boron target power can increase the content of sp^2^ (C=C) (Figure 4b,d,f). And the I_D_/I_G_ is increased with increasing both boron and carbon target powers (Table 1, sample 1,2,3,4). So, the high target power decreasing the E_opt_ is due to an increasing in the content of sp^2^. It is worth mentioning that a higher carbon target power doesn’t generate B–C bonds (Table 1, sample 4). The reason for this may be that high energy carbon atoms only promote the reaction between carbon atoms, and lower boron target power can’t supply enough energy to form B–C bonds. Moreover, this B–C bond may be due to the E_opt_ of 2.78 eV, as shown in Figure 3b, which indicates that B–C bonds further reduce the E_opt_. With these results, the most important factor in the modulation of E_opt_ in a-C:B is the supplied energy. So, boron atoms can induce the formation of sp^2^ hybridization in a-C:B with sufficient energy. The narrow E_opt_ is preferable in a-C films, as reported, and the a-C can be used in solar cells and semiconductors [12]. The narrow E_opt_ benefits from electron transition, which is related to higher performance. Although the minimum E_opt_ of this a-C:B is just 2.78 eV, changing the boron and carbon target powers can further modulate the E_opt_. Thus, this work is significant for the future development of a-C.

Thus, the narrowing of the E_opt_ is mainly attributed to the increase in sp^2^ content. This is consistent with previous reports [22]. However, the content of boron may not the most important factor in the modulation of E_opt_, and the supplied energy is the crucial to the generation of sp^2^ hybridization. We propose the following reasons for this narrowing E_opt_: (a) with lower energy, the boron atoms only act on weakly π bonds of sp^2^ hybridization in carbon atoms; hence, boron atoms can’t react with carbon atoms, and boron atoms act as mixture in a-C:B films. However, the mixture of boron atoms can also serve as a fluxing agent for carbon to form sp^2^ clusters due to the fact that boron has a lower melting point (about 2075 °C) compared to carbon (3550 °C). Thus, boron atoms can still promote the formation of sp^2^ clusters with carbon; (b) with higher energy, the boron atoms are able to act on σ bonds in sp^3^ hybridization to form B–C bonds with terminate edges, and can also induce sp^3^ hybridization transfer to sp^2^ hybridization. The intrinsic reason for this is that the π bonds are the weakest, and σ bonds in sp^2^ hybridization are stronger than σ bonds in sp^3^ hybridization; thus, boron atoms can influence different carbon bonds with different energies.

## 4. Conclusions

In this work, variations of E_opt_ were obtained in a-C:B films. Increasing boron and carbon target powers can increase the sp^2^ content to narrow the E_opt_ from 3.19 eV to 2.78 eV. The content of boron may not the most important factor in the modulation of the E_opt_, but the supplied energy is crucial in generating sp^2^ hybridization. So, these results show that boron atoms can act as a fluxing agent to promote the formation of sp^2^ clusters at lower energy with carbon. Also, boron atoms prefer to act on carbon σ bonds in sp^3^ hybridization to induce a sp^3^ hybridization transfer to sp^2^ hybridization under higher energy. This work is significant to advances the present understanding of a-C and the preparation of new amorphous films.

## Figures and Tables

**Figure 1 materials-12-01780-f001:**
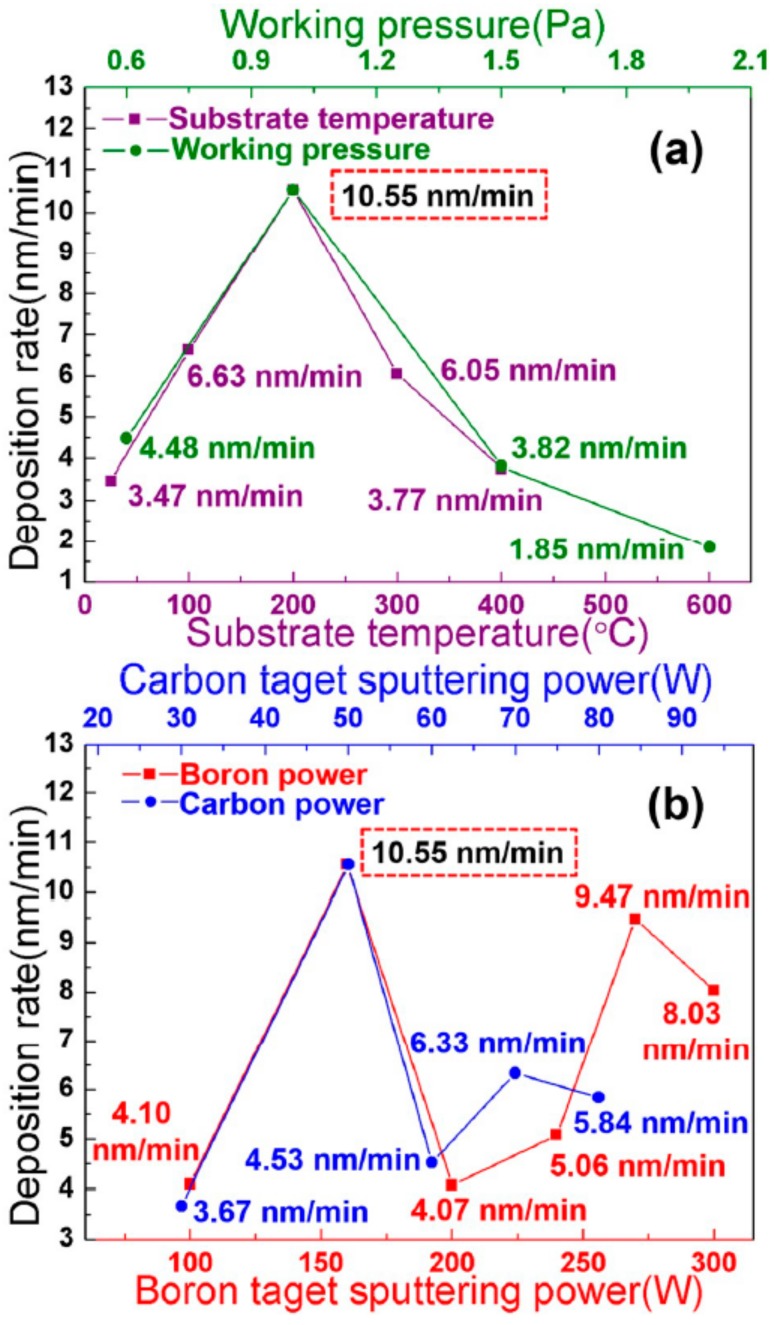
The deposition rate patterns; (**a**) is the deposition rate relates to substrate temperature (purple) and working pressure (green); (**b**) is the deposition rate relates to boron target power (red) and carbon target power (blue). The rate is 10.55 nm/min with optimal conditions, i.e., 200 °C, 1.0 Pa, boron power 160 W, and carbon power 50 W.

**Figure 2 materials-12-01780-f002:**
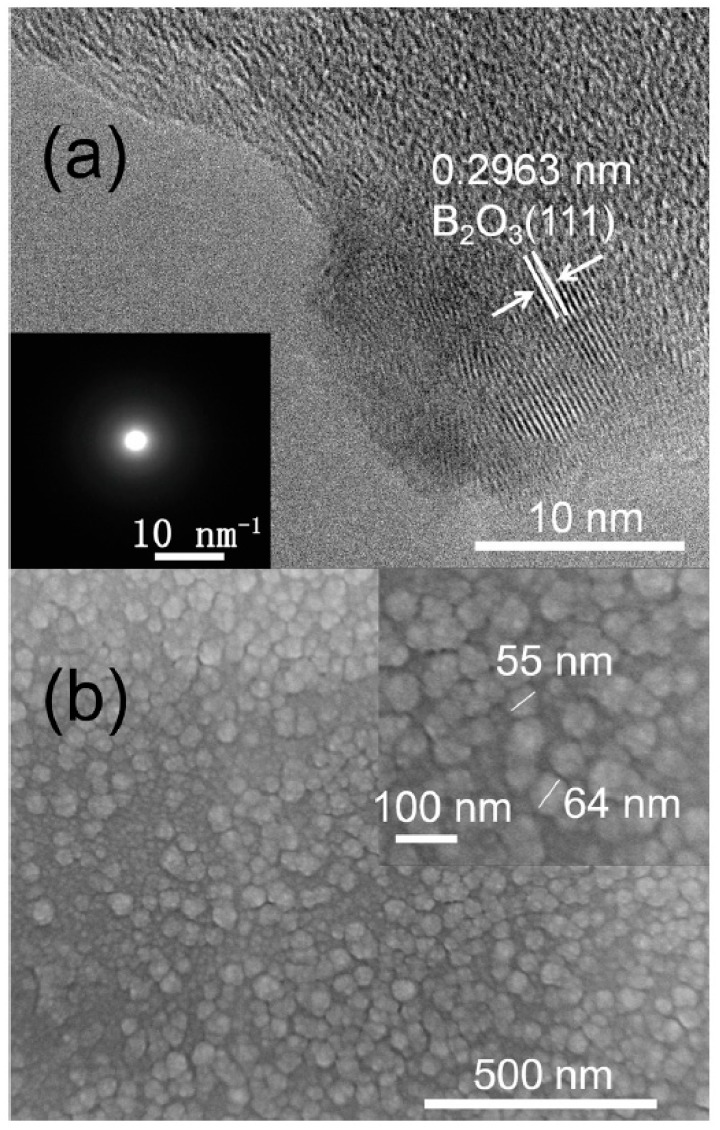
The microstructure and morphology of a-C:B (200 °C, 1.0 Pa, boron target power 200 W, and carbon target power 50 W). (**a**) is a high resolution transmission electron microscopy (HRTEM) pattern; the inset pattern in (**a**) is the selected area electron diffraction (SAED) results. (**b**) is a scanning electron microscopy (SEM) pattern. HRTEM and SAED indicate a-C:B is amorphous; there is B_2_O_3_ on the surface of a-C:B. SEM pattern, indicating that a-C:B is granulated with a nano size of about 30–80 nm.

**Figure 3 materials-12-01780-f003:**
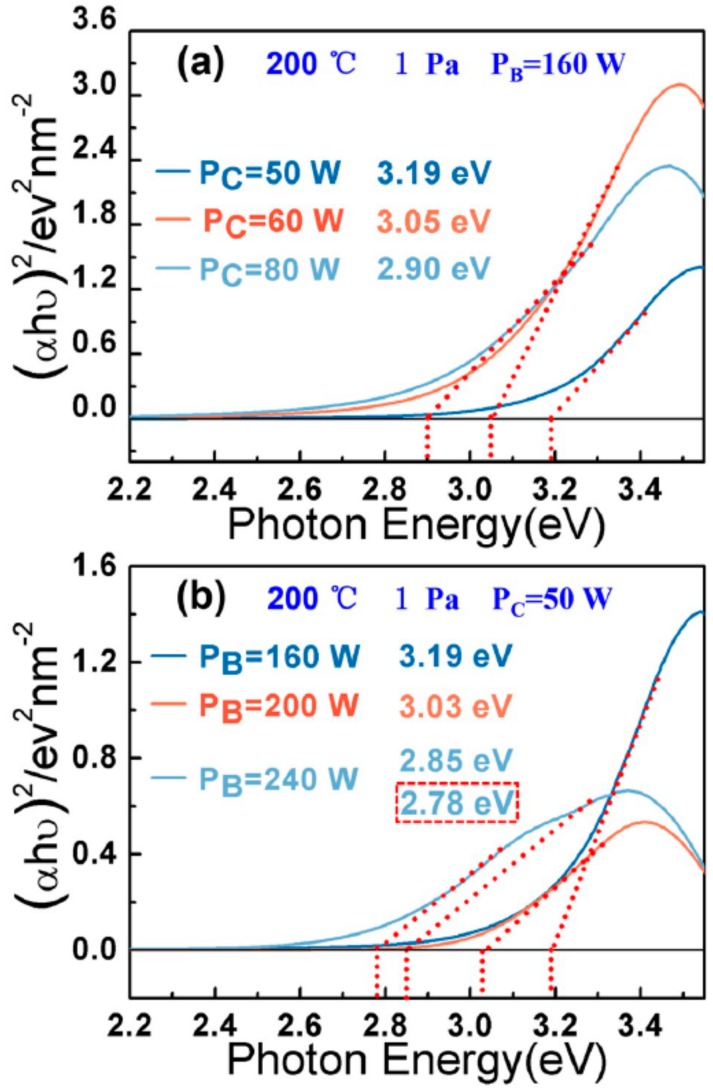
The optical band gap (E_opt_) pattern of a-C:B. (**a**) E_opt_ relates to different carbon target power; E_opt_ decreases with increasing carbon target power. (**b**) E_opt_ relates to different boron target power; the E_opt_ decreases with increasing boron target power.

**Figure 4 materials-12-01780-f004:**
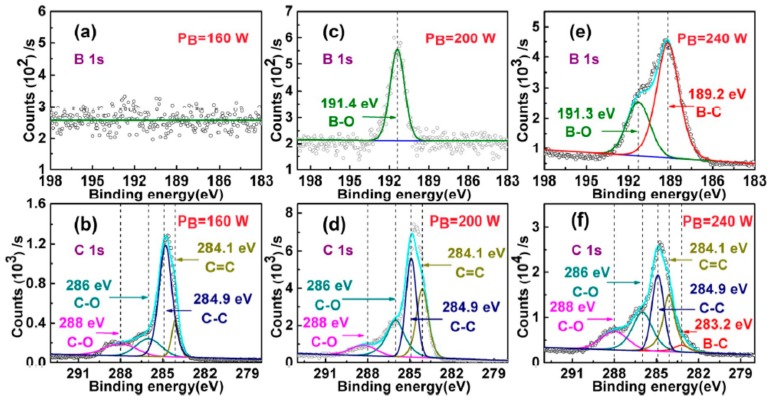
X-ray photoelectron spectroscopy (XPS) patterns relate to different boron target power. All samples are synthesized under the same conditions, i.e., 200 °C, 1.0 Pa, and carbon target power of 50 W. (**a**,**b**) are XPS results under boron target power of 160 W. (**c**,**d**) are XPS results under boron target power of 200 W. (**e**,**f**) are XPS results under boron target power of 240 W. (**a**,**c**,**e**) are the boron 1s state. (**b**,**d**,**f**) are the carbon 1s state. With increasing boron target power to 200 W, B–O bonds appear; with increasing boron target power to 240 W, B–C bonds appear; the C–C peak is decreased and C=C is increased with increasing the boron target power.

**Table 1 materials-12-01780-t001:** Synthesis parameters and tested results of samples. All target-substrate distances and deposition times were 55 nm and 30 min respectively; boron target power (P_B_), the unit is W; carbon target power (P_C_), the unit is W; substrate temperature (T), the unit is °C; the XPS peak position of binding energy are represent by B–O, C–O, C–C, C=C, B–C, the unit is eV; E_opt_ is the optical band gap, the unit is eV; I_D_/I_G_ is the proportion of D and G modes in the Raman results.

Sample	P_B_	P_C_	T	Pa	B–O	C–O	C=C	C–C	B–C	E_opt_	I_D_/I_G_
1	160	50	200	1.0	-	286; 288	284.1	284.9	-	3.19	57%
2	200	50	200	1.0	191.4	286; 287.8	284.1	284.9	-	3.03	59%
3	240	50	200	1.0	191.3	286; 288	284.1	284.9	189.2; 283.2	2.78	61%
4	160	80	200	1.0	191.3	286; 288	284.1	284.9	-	2.90	64%
5	160	50	200	2.0	-	286.2; 288.2	284.1	284.9	-	3.08	57%
6	160	50	300	1.0	191.4	286; 288	284.1	284.9	189.2; 283.2	3.16	59%
7	160	50	400	1.0	191.2	286; 288	284.1	284.9	189.2; 283.2	3.15	61%

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
