# Peer review of "Modulating Band Gap of Boron Doping in Amorphous Carbon Nano-Film"

_materials, 2019, doi:10.3390/ma12111780_

Reviewer 1 Report

The abstract needs to be improved especially in terms of English language.

Please increase the resolution of Fig.1. The Fig.1 also needs to have larger font size to be easier to read.

Please don't use abbreviation in Fig.2 caption.

Please also increase resolution of Fig.3.

There are too many tiny words in Fig.4 and hard to read. Please reorganize the images so it is easier to read. The caption of the figure must also mention what is the main key point of the figure. The figure caption should be complete.

Please improve the manuscript as you think what would be the best to attract readers and to help readers to understand the science and engineering of boron doping.

Author Response

Response to Reviewer 1 Comments

Point 1: The abstract needs to be improved especially in terms of English language.

Respond 1:

The abstract has carefully revised, and the English language has revised for whole article, we hope this type is more readable.  

Point 2:Please increase the resolution of Fig.1. The Fig.1 also needs to have larger font size to be easier to read.

Respond 2:

    We are so sorry that our pictures are not clear. As review’s suggestion, the resolution of Figure 1 has improved, and the four sub-graphs have changed to two sub-graphs with single column, thus the Figure 1 has larger font size and more easy to read. And Figure 1’s caption has revised too.

Point 3:Please don't use abbreviation in Fig.2 caption.

Respond 3:

    Thanks for reviewer’s reminding. We have revised the Figure 2’s caption without abbreviation, and the Figure 2 has revised with higher resolution and the font size is more clearly.

Point 4:Please also increase resolution of Fig.3.

Respond 4:

    We are so sorry that our pictures are not clear. As review’s suggestion, the resolution of Figure 3 has improved, and the four sub-graphs have changed to two sub-graphs with single column, and the Figure 3’s caption has revised too. We hope it will more clearly than before.

Point 5:There are too many tiny words in Fig.4 and hard to read. Please reorganize the images so it is easier to read. The caption of the figure must also mention what is the main key point of the figure. The figure caption should be complete.

Respond 5:

We are so sorry for that Figure 4 is look like complicate and the words in it is not clear. As review’s suggestion, the Figure 4 has shrink to six sub-graphs, only maintain the XPS results of boron target power change, which is most important part to indicate the reason of Eopt narrowed (proportion of sp2/sp3). And other XPS peak results are put into Table 1. And the Figure 4’s caption has also revised. We hope it will more clearly than before.

Point 6: Please improve the manuscript as you think what would be the best to attract readers and to help readers to understand the science and engineering of boron doping.

Respond 6:

    Thanks for review’s suggestion, we are so sorry for that we hadn’t expressed our view clearly. We have carefully improved the manuscript, and revised the analysis of Eopt change, and hope it is better. In this work, we focus on exploring the reason of Eopt change in a-C:B films. And in our conception, we uncovered that the boron content is not the most important factor to modulate Eopt, but the supplied energy is important. The boron atoms only as a mixture in a-C:B with low energy. But the boron atoms can take effect as fluxing agent due to lower melting point than carbon, and boron atoms can promote carbon atoms to form sp2 hybridization. Moreover, boron atoms can break σ bonds in carbon sp3 hybridization to for B-C bonds with increasing the content of sp2 hybridization under enough energy. Thus boron atoms can induce the carbon sp3 hybridization transfer to sp2 hybridization with high energies. This is the science and engineering of boron doping in this work. 

Reviewer 2 Report

This is a concise study where the authors report the preparation of B doped a-C (a-C:B) nano-films by magnetron sputtering. The purpose is to explore the variations the optical band gap of the film as a function of the fabrication parameters. Different experimental conditions, such as increasing boron target power, carbon target power, substrate temperature, and working pressure are considered.  The authors focus on the conversion of sp3 into sp2 bonding induced by boron atoms at higher energies.

The topic is interesting and the paper contains possibly useful information for the fabrication of a-C and other amorphous films. In my opinion, the paper can be considered for the publication after a moderate revision.  

Here are my suggestions:

-          English grammar and style must be improved.

-          “However, when further increasing the sputtering power, the deposition rate is decreased attribute to increase the anti–splash rate of the deposition atoms.  Thus the undulance of deposition rate in high power may attribute to the balance of the ionization rate of the argon and the anti–splash rate of the deposition atoms.” What is anti-splash? What is the physical mechanism behind it?

-          All figures must be magnified. Figure 4 is really difficult to read. Make the plots in Figure 4 bigger and try to organize them in a 3x3 (or 3x4) array.

-          The main point of the paper is that boron doping can be controlled during the fabrication process to reduce the optical bandgap. The authors should highlight the advantages of having a slightly reduced optical bandgap in a:C.

Author Response

Response to Reviewer 2 Comments

Comments:

This is a concise study where the authors report the preparation of B doped a-C (a-C:B) nano-films by magnetron sputtering. The purpose is to explore the variations the optical band gap of the film as a function of the fabrication parameters. Different experimental conditions, such as increasing boron target power, carbon target power, substrate temperature, and working pressure are considered. The authors focus on the conversion of sp3 into sp2 bonding induced by boron atoms at higher energies.

The topic is interesting and the paper contains possibly useful information for the fabrication of a-C and other amorphous films. In my opinion, the paper can be considered for the publication after a moderate revision.  

Here are my suggestions:

Point 1:English grammar and style must be improved.

Respond 1:

    Thanks for review’s advice. The English language has revised for whole article, we hope this type is more readable.

Point 2:“However, when further increasing the sputtering power, the deposition rate is decreased attribute to increase the anti–splash rate of the deposition atoms.  Thus the undulance of deposition rate in high power may attribute to the balance of the ionization rate of the argon and the anti–splash rate of the deposition atoms.” What is anti-splash? What is the physical mechanism behind it?

Respond 2

We are so sorry for that it is a wrong word of “anti-splash”, the correct word is “re-sputtering”, we have revised the wrong word of “anti-splash” to “re-sputtering” in our article. “Re-sputtering” is a phenomenon at high sputtering power. Such as in boron target, in general, boron ions have enough energy to bombard the substrate, and boron atoms will growth on substrate, the deposition rate will be increased with increasing boron target power. But when the boron target power is too high, the boron ions will have too much energy to bombard the substrate, the samples on the substrate will be bombarded, and break away from substrate, thus the deposition rate will be decreased with increasing boron target power at this situation.

Point 3:All figures must be magnified. Figure 4 is really difficult to read. Make the plots in Figure 4 bigger and try to organize them in a 3x3 (or 3x4) array.

Respond 3:

Thanks for review’s reminding, we are sorry for that the Figures are not clearly and difficult to read. We have revised all Figures. Figures resolution have improved; Figures font is lager; four sub-graphs in Figure 1 and Figure 3 have shrink to two sub-graphs, respectively; Ten sub-graphs in Figure 4 has change to six sub-graghs. We hope it will be more clearly and readable. 

Point 4:The main point of the paper is that boron doping can be controlled during the fabrication process to reduce the optical bandgap. The authors should highlight the advantages of having a slightly reduced optical bandgap in a:C.

Respond 4:

    Thanks for review’s reminding, we have added the advantage of a-C:B films in in line 373-377, page 6. And our work found a way to modulate the Eopt of a-C:B, Although the minimum Eopt of these a-C:B is just 2.78 eV, but change the boron target power and carbon target power can further modulate the Eopt. So our work is meaningful.

Round  2

Reviewer 1 Report

Thank you for answering my questions in the review report.